# Lipopolysaccharide of the *Yersinia pseudotuberculosis* Complex

**DOI:** 10.3390/biom11101410

**Published:** 2021-09-26

**Authors:** Yuriy A. Knirel, Andrey P. Anisimov, Angelina A. Kislichkina, Anna N. Kondakova, Olga V. Bystrova, Anastasia S. Vagaiskaya, Konstantin Y. Shatalin, Alexander S. Shashkov, Svetlana V. Dentovskaya

**Affiliations:** 1Laboratory of Carbohydrate Chemistry, N. D. Zelinsky Institute of Organic Chemistry, Russian Academy of Sciences, 119991 Moscow, Russia; annakond@gmail.com (A.N.K.); bystrova@interlab.ru (O.V.B.); shash@ioc.ac.ru (A.S.S.); 2Laboratory for Plague Microbiology, Especially Dangerous Infections Department, State Research Center for Applied Microbiology and Biotechnology, 142279 Obolensk, Russia; a-p-anisimov@yandex.ru (A.P.A.); angelinakislichkina@yandex.ru (A.A.K.); vagaiskaya.anastasiya@gmail.com (A.S.V.); 3Department of Biochemistry and Molecular Pharmacology, New York University School of Medicine, New York, NY 10016, USA; kshatalin@yahoo.com

**Keywords:** *Yersinia pseudotuberculosis*, *Yersinia pestis*, lipopolysaccharide (LPS), lipid A, core, pathogenicity factor, pathogenesis, plague

## Abstract

Lipopolysaccharide (LPS), localized in the outer leaflet of the outer membrane, serves as the major surface component of the Gram-negative bacterial cell envelope responsible for the activation of the host’s innate immune system. Variations of the LPS structure utilized by Gram-negative bacteria promote survival by providing resistance to components of the innate immune system and preventing recognition by TLR4. This review summarizes studies of the biosynthesis of Yersinia pseudotuberculosis complex LPSs, and the roles of their structural components in molecular mechanisms of yersiniae pathogenesis and immunogenesis.

## 1. Introduction

The lipopolysaccharide (LPS) is the major constituent of the outer leaflet of the outer membrane of Gram-negative bacteria, including the genus *Yersinia*. Its lipid moiety called lipid A is embedded in the membrane and serves as an anchor for the rest of the LPS molecule. The S (smooth)-form LPS possesses an outermost repetitive glycan region designated as the O-specific polysaccharide (O-polysaccharide, OPS) as it defines the serospecificity of the bacteria and, therefore, is often called O-antigen. The OPS is linked to lipid A via a negatively charged oligosaccharide called core. Both the lipid A and core are important for the integrity and regulation of the permeability of the outer membrane. The R (rough)-form LPS is limited to the core and lipid moieties. The OPS is a highly variable portion of the LPS and is used as the basis for bacterial O-serotyping. It also provides protection to the microorganisms from host defense mechanisms, such as complement-mediated killing and phagocytosis.

Currently, the genus *Yersinia* comprises 27 species, of which several species, including *Yersinia pseudotuberculosis*/*Yersinia pestis*, *Yersinia similis* and *Yersinia wautersii* (or the Korean group of strains), are combined into the so-called *Y. pseudotuberculosis* complex. Except for *Yersinia pestis*, which has an R-form LPS, the other members of the *Y. pseudotuberculosis* complex are able to produce an S-form LPS. Recent advances in chemical analytical methods, spectrometry and whole-genome sequencing have made it possible to determine the spectra of polymorphisms of fine structures of the constituent components of *Yersinia* molecules and to begin to determine the functional significance of individual elements of this polyfunctional molecule. This review focuses on recent studies of the LPS structure and genetics of biosynthesis of these molecules and discusses the relationship between the LPS molecular structure and the virulence of the representatives of the *Y. pseudotuberculosis* complex.

## 2. *Y. pestis* and *Y. pseudotuberculosis* Lipopolysaccharide Structures

### 2.1. Composition and Structure of the O-Specific Polysaccharides

Currently, the O-serotyping scheme of *Y. pseudotuberculosis* [1], which may also be applied to the other species of the complex [2,3], includes 21 O-serotypes, of which, only 18 have been validated as the O-antigen-based serotypes. The other three, O8, O13 and O14, are expressed by rough mutants and define no O-specificity. Therefore, they must be excluded from the O-serotyping scheme [4].

Smooth strains of *Y. pseudotuberculosis* have branched regular OPSs made up of tetra- or penta-saccharide O units with a di- to tetra-saccharide repeat in the main chain and a mono- or di-saccharide side chain(s) (Table 1). The OPSs are synthesized by the polymerase-dependent pathway, which includes the assembly of the O unit on a lipid carrier at the periplasmic side of the inner membrane followed by polymerization. Either 2-acetamido-2-deoxyglucose (GlcNAc) or 2-acetamido-2-deoxygalactose (GalNAc) has been demonstrated to be the first monosaccharide of the O unit [5], whose transfer to the lipid carrier initiates the synthesis of the serotypes O4b and O3 OPSs, respectively. Genetic data suggest that an amino sugar (GlcNAc or GalNAc) is the first in the O units of the other O-serotypes too, as shown in Table 1.

In most O-serotypes, main-chain components are rather common monosaccharides, such as d-Glc, d-Gal, d-Man, l-Fuc, d-GlcNAc and d-GalNAc, which all exist in the pyranose form. The exceptions are the OPSs of serotypes O12 and O9, which contain rarely occurring deoxy sugars, 6-deoxy-l-glucose (l-Qui) and *N*-acetimidoylamino-2,6-dideoxy-l-galactose (l-FucNAm), respectively, in the main chain. The latter is the only acidic OPS of *Y. pseudotuberculosis* due to the presence of 3-*O*-acetylated 2-acetamido-2-deoxy-d-glucuronic acid (d-GlcNAcA).

In contrast to the main chains, the side chains in most OPSs are composed of unusual monosaccharides, such as 3,6-dideoxyhexoses, including abequose (Abe), ascarylose (Asc), colitose (Col) and tyvelose (Tyv), which are always pyranosidic, as well as paratose (Par), which occurs as either a pyranoside or furanoside. Abequose with a 1-hydroxyethyl side chain called yersiniose A (Yer) occurs in the OPSs of serotypes O6 and O12. Two more unusual side-chain deoxy sugars, 6-deoxy-l-altrose in the furanose form and 6-deoxy-d-*manno*-heptose (6dmanHep) in the pyranose form, are present in the OPSs of several O-serotypes. The OPSs of serotypes O6 and O7 exceptionally bear a d-glucose or d-galactose side chain, respectively.

It is worth noting that the O-antigen expression by *Y. pseudotubrculosis* is downregulated at 37 °C, resulting in the production of predominantly SR-form LPS, in which the OPS is limited to the single O-unit [5,6]. Various minor modifications are observed to the O unit in the SR-form LPS. When the synthesis of 6dmanHep is impaired, its biosynthetic precursor, d-*glycero*-d-*manno*-heptose, is incorporated into the O-unit of *Y. pseudotuberculosis* O2a in place of 6dmanHep [7]. Minor hexosylation (presumably glucosylation) of the O unit occurs in the SR-form LPS of *Y. pseudotuberculosis* O3 [5].

Many OPSs of *Y. pseudotuberculosis* are structurally and, as a result, serologically related due to the presence of the same lateral monosaccharide, e.g., β-Par*f*, α-Abe*p* or α-Tyv*p* in serotypes O1, O2 and O4, respectively, or α-Col, in serotypes O6, O7 and O10. Some other OPS groups (those of serotypes O1a, O2a and O4b; O1c, O2b and O3; O1b and O11; O2c and O4a; O5a, O5b and O15) share their main chain structure.

Although most OPS structures are unique to *Y. pseudotuberculosis*, there are two exceptions. The OPS of serotype O10 is closely related to that shared by *Escherichia coli* O111 and *Salmonella enterica* O35 [8,9], both differing in the presence of a d-GalNAc residue in the main chain in place of a d-GlcNAc residue. The main chain of the OPSs of *Y. pseudotuberculosis* O2c and O4a is identical to the linear OPS of *S. enterica* O18 [10].

**Table 1 biomolecules-11-01410-t001:** Structures of the O-specific polysaccharides of *Y. pseudotuberculosis*.

Serotype [Reference]	Structure of the Repeating Unit
O1a R = β-Par*f* [11]O2a R = α-Abe*p* [7]O4b R = α-Tyv*p* [12]	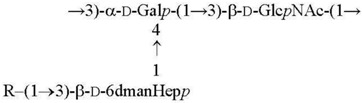
O1c R = β-Par*f* [13]O2b R = α-Abe*p* [14]	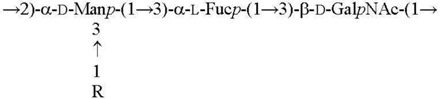
O3 [15]	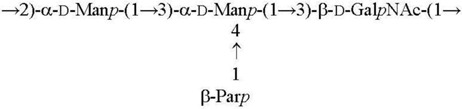
O2c R = α-Abe*p* [15]O4a R = α-Tyv*p* [16]	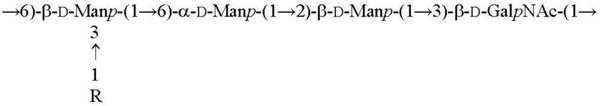
O1b R = β-Par*f* [17]O11 R = α-l-6dAlt*f* [18]	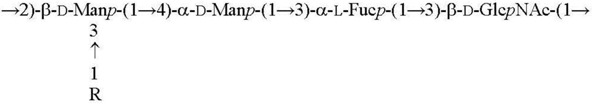
O5a R = α-Asc*p* [19]O5b R = α-l-6dAlt*f* [20]O15 R = β-Par*f* [21]	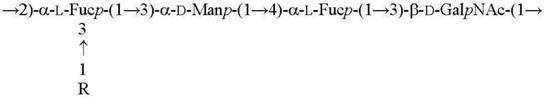
O6 [22,23]	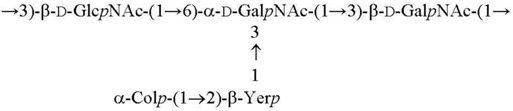
O7 [24]	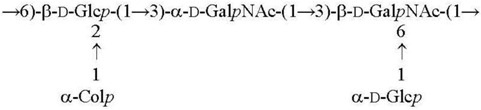
O10 [25]	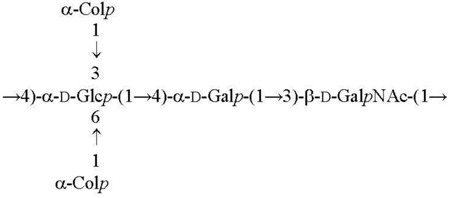
O12 [3]	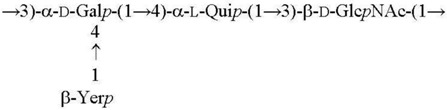
O9 [26,27]	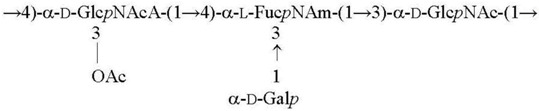

### 2.2. Structural Variants of the Core Region

The structure of the inner core consisting of two residues of oct-2-ulosonic acids (Kdo or Ko) and three residues of l-*glycero*-d-*manno*-heptose (Hep) is conserved in the Enterobacteriales order, including the genus *Yersinia*.

The first heptose residue linked to Kdo carries a β-d-Glc residue, which is characteristic of the so-called non-S*almonella* core type (Table 2). In the R-form LPS of *Y. pseudotuberculosis* and LPS of *Y. pestis,* a β-d-GlcNAc residue is present at position three of the second heptose residue in a non-stoichiometric amount. This position is the site of attachment of a long-chain OPS or a single O unit in the S- and SR-form LPS of *Y. pseudotuberculosis,* respectively.

When *Y. pestis* ssp. *pestis*, the main subspecies of *Y. pestis*, is cultivated at 37 °C, which mimics the condition in not hibernating mammals, position seven of the third heptose residue is occupied by a residue of d-*glycero*-d-*manno*-heptose (dd-Hep) (Table 2). The cultivation of bacteria at lower temperatures resulted in two significant changes in the core, including the replacement of dd-Hep with d-Gal and 3-hydroxylation of the sidechain Kdo residue to afford a Ko residue [5,28] (Table 2). Almost full replacement is observed at 6 °C, whereas at 25 °C, mainly mixed core glycoforms, dd-Hep+Ko and d-Gal+Kdo, are produced. A similar temperature-dependent alternation of these two pairs of the lateral monosaccharides also occurs in the core of *Y. pseudotuberculosis* [5]. A significant distinction of *Y. pestis* ssp. *microti* bvv. caucasica and altaica is the inability to incorporate dd-Hep into the core at any temperatures [5] (Table 2).

The incorporation of Gal into the core, but not the hydroxylation of Kdo, is under the control of the two-component PhoPQ signal transduction system [30].

The content of GlcNAc increases, and that of the glycine present at an unknown position in the core of *Y. pestis* decreases with a growth temperature elevation [5,31].

When cultivated at 6 °C, *Y. pestis* also produces another LPS variant, which is distinguished by the phosphorylation of the Ko (major) or Kdo (minor) residue with 2-aminoethyl phosphate (PEtN) combined with the presence of dd-Hep [29] (Table 2). Minor PEtN is also present at 25 °C on the Ko residue in the core of *Y. pestis* bv. altaica but not ssp. *pestis*.

### 2.3. Structure Variations of Lipid A

As with that of the core, the lipid A structure of *Y. pestis* was studied in detail. At 25 °C, the bacteria produce a hexaacylated bisphosphorylated lipid A species containing four primary 3-hydroxymyristoyl (3HO14:0) groups and secondary lauroyl 12:0 and palmitoleoyl 16:1 groups. Each of the phosphate groups carries a 4-amino-4-deoxy-l-arabinose (Ara4N) residue (Figure 1A). This hexaacyl lipid A is accompanied by a pentaacyl species that lacks the 16:1 group (Figure 1B) and a tetraacyl species that is devoid of both secondary acyl groups (Figure 1C).

An alternative pentaacylated lipid A that is distinguished by the presence of four primary 3HO14:0 groups and the secondary 16:1 group (Figure 1D), occurs in *Y. pestis* cultivated at 6 °C [29]. Remarkably, at this temperature, Ara4N is absent from some lipid A species [29].

Upon cultivation at an elevated temperature, the acylation degree and the content of Ara4N significantly decrease so that at 37 °C, the lipid A of *Y. pestis* consists mainly of tetraacyl species with four primary 3HO14:0 groups and non-glycosylated phosphate groups. There are also minor triacyl species and those containing a diphosphate group.

At 37 °C, lipid A of *Y. pseudotuberculosis* lacks diphosphate-containing species, and the content of Ara4N is significantly higher than in *Y. pestis*. This lipid A is distinguished by the presence in some species of a secondary 16:0 acyl group [5,32,33].

## 3. Genetics and Biosynthesis

The biosynthesis of lipid A has been studied in detail for enteric bacteria [34], particularly *E. coli*, and this pathway is assumed to be generally conserved in members of the *Y. pseudotuberculosis* complex. The pathway is mediated by nine enzymes (Table 3) and located in the cytoplasm and on the inner surface of inner membrane. All of the corresponding homologous genes encoding enzymes required for the biosynthesis of the biphosphorylated precursor of enterobacterial lipid A (lipid IV_A_) structure (including the primary acyltransferases LpxA and LpxD, the deacetylase LpxC, the nucleotidase LpxH, the disaccharide synthase LpxB and the kinase LpxK) were identified in the genome of *Y. pestis* (Table 3). The minimal LPS structure needed for the viability of *Y. pestis* is lipid IV_A_, although such *Y. pestis* mutants exhibit highly attenuated growth [35].

Different forms of lipid A could then be synthesized and modified from lipid IV_A_ by different enzymes depending on the environmental conditions. “Late” acyltransferases responsible for attaching secondary acyl substituents, such as myristoyl and palmitoleyl, are encoded by *Y. pestis lpxM* and *lpxP* genes, respectively. The *Y. pestis* LpxM and LpxP transfer laurate and palmitoleate to 3-hydroxymyristate residues at positions 3′ and 2′ GlcNII, respectively. *E. coli* gene coding for secondary acyltransferase LpxL (HtrB) has not been identified in the *Y. pestis* genome. As with *E. coli*, the addition of the first core sugar 3-deoxy-D-*manno*-oct-2-ulosonic acid (Kdo) to the lipid IV_A_ is strictly necessary for the attachment of the secondary acyl groups in *Y. pestis* [35,36,37,38]. Their expression level increases as the cultivation temperature decreases from 37 to 21 °C. The mutant at both genes synthesizes tetraacyl lipid A, which is similar to lipid IV_A_ (Figure 1C) and identical to that synthesized by wild-type *Y. pestis* strains at 37 °C [37]. Kdo-independent late acylation in the lipid A has been shown in *E*. *coli* under slow growth conditions at low temperatures or upon the overexpression of genes encoding late-acyl transferases [39,40,41].

The outer membrane palmitoyl transferase protein PagP was originally identified in *Salmonella* and is responsible for the addition of the secondary 16:0 acyl group (palmitate) to lipid A [42]. PagP activity provides resistance to certain CAMPs; presumably, increased acyl-chain packing prevents the translocation of CAMPs across the outer membrane [42]. The addition of a palmitate to lipid A by enzyme PagP also occurs in immunostimulatory *Y. pseudotubrculosis* and *Y. enterocolitica* strains, but not in immune-evasive *Y. pestis* [5,32,43]. The analysis of *Y. pestis pagP* gene sequences identified a single-nucleotide polymorphism that results in a premature stop in translation, yielding a truncated, non-functional enzyme [36].

ArnT is an amino-arabinose transferase that transfers L-Arn4N to the 4′-phosphate of lipid A. The biosynthesis pathway of aminoarabinose and its addition to lipid A has been well characterized in *E. coli* and *S. enterica* (reviewed by Trent [44] and Bishop [36]). A complete inner core (Kdo, HepI, HepII, Glc and HepIII) is required to ensure the most efficient incorporation of Ara4N, whereas the presence or absence of core monosaccharides distal from lipid A (GlcNAc, Gal and dd-Hep) has no effect on this process [35,45,46]. As in *E. coli* and *S*. *enterica*, it was found that the *arn* operon in *Y. pestis* is regulated by two-component signal transduction systems, PhoP/PhoQ and PmrA/PmrB [32,47].

A homologous gene of phosphatase LpxT (YeiU) transferring phosphate from UndPP to lipid A giving rise to diphosphate has also been found in the genome of *Y. pestis* [35,42].

The core biosynthesis genes in *E. coli*, *S. enterica* and a number of other enterobacteria are clustered in a chromosomal region, forming the *waa* cluster [34]. Two clusters (*waaI* and *waaII*) with four and two homologues of the *waa* genes and one cluster with two *wab* genes, which also encode core biosynthetic enzymes, have been identified in the *Y. pestis* genome [35,45,46]. The respective function of each of these proteins (either demonstrated or proposed based on homology comparisons) has been summarized (Table 4).

In *Y. pseudotuberculosis*/*Y. pestis*, the *waaI* cluster contains five genes *gmhD*-*waaF*-*waaC*-*waaA*-*waaE* that are required for the biosynthesis of inner core oligosaccharides (Table 4). The *gmhD*, *waaC* and *waaF* genes encode proteins involved in the biosynthesis and transfer of HepI and HepII, whereas the *waaE* gene is responsible for transferring a Glc residue to the HepI residue in the core. The gene *waaA* encodes WaaA that can add one or two Kdo residues to the lipid IV_A_. The *wab* cluster contains two genes that code for enzymes that are responsible for the biosynthesis of outer core oligosaccharides. The *waaII* cluster contains two genes, *waaQ* and *waaL*. The *waaL* gene encodes a ligase enzyme required for the attachment of O-antigen to the core-lipid A. The heptosyltransferase WaaQ adds a HepIII residue to the HepII [35].

As mentioned above, the terminal Kdo residue in the core at a low temperature is partially substituted for the Ko residue. The latter is synthesized via the oxidation of the 3-deoxy group of Kdo with a unique Fe^2+^/α-ketoglutarate/O_2_-dependent Kdo-3-hydroxylase (KdoO) [48]. KdoO is an inner core assembly enzyme that functions after the Kdo-transferase KdtA but before the heptosyl-transferase WaaC enzyme during the Ko-containing LPS biosynthesis [49].

Gene homologues of transferase EptB (YhjW) transferring PEtN from phosphatidylethanolamine to Kdo [51] have also been found in the *Y. pestis* genome [35].

The structures of core oligosaccharide and lipid A among *Y. pestis* and *Y. pseudotuberculosis* strains are relatively conserved with no strain-to-strain variability in sugar composition. This observation is consistent with the discovery that the genes in the loci for core and lipid A biosynthesis are well conserved among *Y. pestis* and *Y. pseudotuberculosis* strains whose genomes have been sequenced, apart from the absence of the dd-Hep residue as a terminal sugar in the core of *Y. pestis* ssp. *microti* bvv. caucasica, hissarica, xilingolensis, talassica and altaica (Figure 2). Analysis of the *wabC* nucleotide sequence reveals differences among the *Y. pestis* isolates that result in amino acid variation in the polypeptide chain. The following two variable sites in the gene can be identified: (1) insertion of a guanosine at a position of 544 leading to the formation of a terminating codon, which results in the shortening of the polypeptide chain from 326 to 192 a.o. (bv. caucasica); (2) deletion of 14 nucleotides at a position of 84 also leading to the formation of a terminating codon, which results in the shortening of the polypeptide chain from 326 to 35 a.o. (bvv. altaica, xilingolensis, qinghaiensis, hissarica and talassica).

The genetics and evolution of *Y. pseudotuberculosis* complex O-specific polysaccharides have been reviewed in great detail by Kenyon et al. [4]. All the *Y. pseudotuberculosis* complex OPS gene clusters are located between conserved genes *hemH* and *gsk*. The gene cluster generally encodes the enzymes for NDP-sugar synthesis, the glycosyltransferases and polymerases needed for the assembly of the O-antigen. Genes specific to many of the *Y. pseudotuberculosis* O-antigen serotypes were identified in other species in the *Y. pseudotuberculosis* complex [2,53]. The serotyping of *Y. pseudotuberculosis* was originally developed using immunochemical assays [54] but has now been supplemented with genetic methods, such as PCR and sequencing [55]. The O-antigen serotyping scheme should apply to all the members of this complex [2].

*Y. pestis* isolates possess 17 genes similar to those in the *Y. pseudotuberculosis* O:1b OPS gene cluster, and five of them are inactivated by insertions or deletions, suggesting that the plague pathogen originated from its ancestor *Y. pseudotuberculosis* O:1b [56], having lost the ability to form O polysaccharide during speciation [6,55,57]. Furthermore, 16 out of 17 genes of the two bacteria in O-antigen biosynthesis clusters are 99–100% identical, while the *wzx* gene encoding flippase is only 90.4% identical [4]. It is assumed that, in contrast to *Y. pseudotuberculosis* Wzx, which guarantees the transmembrane transfer of the UndPP-bound pentasaccharide repeating unit of the O-antigen, the flippase of *Y. pestis* is redirected to the transfer of one UndPP-bound GlcNAc residue across the inner membrane, which subsequently binds with the LPS core using WaaL ligase in the same position as the O polysaccharide in *Y. pseudotuberculosis*.

## 4. Biological Effects of *Y. pestis* and *Y. pseudotuberculosis* Lipopolysaccharide

LPS is an integral component of the outer membrane that is in direct contact with the host organism, taking part in the protection of bacterial cells against complement system components and other host defense mechanisms and systems [58,59]. Indeed, identification with the help of the whole-genome signature-tagged mutagenesis of the genes responsible for *Y. pseudotuberculosis* YPIII growth in the mammalian host organism showed that up to a third of the derivatives attenuated in the murine-pseudotuberculosis-infection model carried mutations in the LPS-biosynthesis (mainly LPS core or O-antigen biosynthesis) genes [60,61].

In a more recent study from another lab, the profile of the identified by dint of signature-tagged mutagenesis *Y. pestis* virulence-associated genes did not contain genes involved in LPS biosynthesis at all. The authors of the publication admit [62] that this may have been due to the fact that O-antigen-biosynthesis genes are non-functional in *Y. pestis* [57] and/or to the minor role of LPS in *Y. pestis* pathogenicity. The other reasons for inconsistency might be the use of different animal models and/or different routes of infection or just the limited representation of the library used for screening [62].

The detailed structure of lipopolysaccharide differs not only from one bacterial species to another but even within single species from one bacterial cell to another, which is consistent with the recent discovery of additional enzymes and gene products that can alter the basic structure of LPS adapting bacteria to exist in different environmental conditions, which in the case of pathogenic bacteria are various host niches. These modifications are not required for survival, but they are tightly regulated in the cell and are closely related to bacterial virulence [63].

Taking into account the complexity of the genetics of LPSs synthesis, the large number of enzymes and proteins responsible for the biosynthesis and export of LPSs, the dependence of the LPS structure on the cultivation conditions, as well as the simultaneous presence of several structural variants of LPS in the bacterial cell, one can expect significantly conflicting results in experiments not only carried out in different laboratories, but even in the same laboratory. To avoid this problem, it would be reasonable to assess the contribution of individual structural components of yersiniae LPSs to the pathogenesis of plague infection using standard operating procedures on sets of isogenic mutants generated on the basis of similar if not the same well-characterized “wild-type” strains. To reduce the multicomponent nature of such experiments, it is better to compare only one function of this polyfunctional molecule in one study.

### 4.1. LPS Recognition by Innate Immunity

The LPS of Gram-negative bacteria plays multiple key roles in the interaction between the pathogen and the infected host. First, LPS is among the pathogen-associated molecular patterns (PAMPs) recognized by Toll-like receptors (TLR), the structure and properties of which depend both on the species of the bacterium and on the conditions of its cultivation [64]. The evasion of innate immunity through the decrease in the detectability of pathogen-associated patterns to their complete invisibility underlies the pathogenesis of the plague. During its cyclic alternate existence in organisms of mammalian hosts and insect vectors, *Y. pestis* modifies its outer leaflet of the outer membrane by changing the structure of the lipid A portion of its LPS recognized by TLR4-MD2. Depending on the structure of LPS lipid A, *Y. pestis*, as with any other bacterium, is either recognized by the host organism as a foreign molecular structure of infectious origin that then triggers a cascade of immune responses of optimal strength aimed at eliminating the pathogen (immunostimulatory hexa-acyl LPS), or it evades detection by the immune system (immune-evasive tetra-acyl and penta-acyl LPS), leading to the uncontrolled multiplication of the plague pathogen (septicemia), accompanied by the massive production of LPS with low endotoxic activity, which, when the “critical mass” is reached (endotoxemia), in any case, causes the excessive uncontrolled production of proinflammatory cytokines, including the major mediator of septic shock, tumor necrosis factor α (TNF-α), and ultimately leads to septic shock development [28,32,65,66]. *Y. pestis* LPS stimulated the production of TNF-α and IL-6 from mouse macrophages but was less active in these assays than the LPS isolated from *E. coli* strain 0111 [67].

An agonistic activity of LPS from wild-type *Y. pestis* in relation to TLR4 was compared with that of LPSs from *E. coli* mutants deficient in LpxM and LpxP acyltransferases, which were in charge of the embedding of secondary fatty acid residues (12:0 and 16:1) into lipid A, as well as to LPSs from bacteria of the genus *Psychrobacter* comprising lipid A fatty acids with shorter acyl chains (C10–C12) than those in lipid A from *Y. pestis* or *E. coli* (C12–C16). A dependence of the TNF-α-inducing activity of LPS on the number or length of acyl chains within lipid A was revealed [68].

### 4.2. LPS Interaction with Antimicrobial Peptides (AMPs)—The First Line of Defense in Innate Immunity

Cationic antimicrobial peptides (CAMPs), which are among the classes of AMPs (such as defensins, cathelicidins and kinocidins), the most ancient and efficient components of host defense, play a central role as effector molecules of innate immunity. The majority of them not only hastily kill a wide range of bacteria but also modulate immunity and other host responses [69].

CAMP resistance in the plague pathogen as well as in a number of other Gram-negative bacteria depends on LPS modifications with Ara4N [70] that are believed to camouflage both of the two *Y. pestis* LPS domains, namely, lipid A and core polysaccharide negative charges, and decrease the affinity of positively charged CAMP to the bacterial superficies. A high resistance of the wild-type *Y. pestis* strains with a near-stoichiometric content of Ara4N in the LPS (two Ara4N residues per molecule), which is attained by culturing bacteria at 20–28 °C, has been demonstrated using the polymyxin B model. A decrease in the resistance to CAMP correlates with a noticeable drop in the content of Ara4N as the temperature is increased up to 37 °C [66,71]. Mutants with gene *galU* knockouts encoding the pathway of Ara4N synthesis [72], *arnT* [35,45,46] encoding Ara4N-transferase or *phoP* [30,32] modulating the binding of Ara4N to lipid A are sensitive to CAMP independently of the cultivation temperature. The role of Ara4N is also supported by a marked increase in the content of this monosaccharide in the LPS of bacteria cultivated at 37 °C in the presence of polymyxin B [28]. An increase in the Ara4N content in LPS and, as a result, in the resistance of *Y. pestis* to CAMPs with a decreasing cultivation temperature is undoubtedly of adaptive character. High resistance to polymyxin B at a temperature characteristic of insects can presumably be attributed to a greater contribution of CAMPs to the innate immunity defense mechanisms of insects as compared to that of mammals, which have a complement system in addition to CAMPs. More recently, the crucial role of Ara4N modification in *Y. pestis* LPS in guaranteeing resistance against *Xenopsylla cheopis* cheopin, flea cecropin CAMP, was experimentally proven [73].

Another cationic component of LPS, glycine [28], located in the core, can contribute to a certain extent to the resistance to CAMPs, whereas uncharged core components do not seem to play a significant role. An increase in the susceptibility to polymyxin B, which was observed for a set of knockout mutants at glycosyltransferase genes producing a truncated core could presumably be attributed to the simultaneous decrease in the Ara4N content in lipid A due to the inefficiency of the Ara4N transfer to LPS molecules with an incomplete carbohydrate moiety [29,35,45].

The core oligosaccharides of *Y. pestis* could be modified at very low temperatures (6 °C) by phosphoethanolamine (PEtN) [29,74]. Analogous to Ara4N, the incorporation of PEtN decreases the negative charge on the LPS [75]. It was supposed that PEtN decoration may be accountable for *Y. pestis* resistance to the CAMPs produced by the wintering flea [74].

At 25–28 °C, lipid A containing 3–6 acyl groups is synthesized, while at 37 °C, it is mostly tetraacylated [28,64]. While the tetraacylated form may help to overcome warm-blooded-host immunity, it makes *Y. pestis* more susceptible to the AMP cecropin A [37]. Resistance to AMP, dependent on the structure of lipid A, is not uncommon in other bacterial species as well [39,42]. *Y. pestis* AMP resistance is not only dependent on LPS structural changes, since mutants with an intact LPS structure, but susceptible to AMPs and reduced survival rates in fleas are described [73]. The majority of researchers studying *Y. pestis* AMP resistance have applied PB as a model peptide [28,32,35,46,47,72,76,77,78,79,80,81,82,83,84,85]; however, some *Y. pestis* PB-resistant mutants were not fully resistant to protamine or LL-37, and vice versa, which indicates that resistance to structurally unrelated types of peptides may be dissimilar [79,86].

Antimicrobial chemokines (AMCs) are a recently identified family of peptides defending different organisms from bacterial infection [50]. *Y. pseudotuberculosis* mutants with improved binding to the AMCs CCL28 and CCL25 also had an increased susceptibility to chemokine-dependent cell death. The vast majority of mutants with enhanced binding to AMCs were defective in genes related to LPS biosynthesis. An especially significant influence on susceptibility to AMC-mediated killing was observed in the case of disruption of the *hldD*/*waaF*/*waaC* operon, necessary for ADP-L-glycero-D-manno-heptose synthesis and a complete LPS core oligosaccharide. The periodate oxidation of surface carbohydrates also enhanced AMC binding, while the enzymatic elimination of surface proteins pointedly decreased binding. Altogether, this suggests that the structure of *Y. pseudotuberculosis* LPS significantly affects the antimicrobial potency of AMCs preventing them from interaction with the bacterial cell surface proteins.

### 4.3. Lipopolysaccharide and Yersiniae Virulence

A *Y. pseudotuberculosis* O-polysaccharide-deficient derivative had reduced virulence upon the subcutaneous challenge of mice. The inability to produce LPS with O-side polysaccharide chains alone turned out to be insufficient for the emergence of a hypervirulent phenotype that distinguishes *Y. pestis* from its progenitor, *Y. pseudotuberculosis* [87]. High-density array-based signature-tagged mutagenesis was used to search for novel yersiniae virulence-associated genes. In about a third of attenuated mutants, the transposon was inserted into the genes responsible for the synthesis of the O-antigen or core [60].

The ability to synthesize a stealthy, hypoacylated lipid A structure, which is absent in other Yersiniaceae, arose in *Y. pestis* as a result of the absence of the *lpxL* gene and a mutation in the *pagP* gene, which encode two of the lipid A late acyltransferases [88]. It was shown that *Y. pestis* LPS from bacteria grown at 37 °C can inhibit the stimulation of human monocyte-derived dendritic cells not only via TLR4 signaling but also via TLR2 and TLR3 [89].

Temperature determines the structure of LPS in *Y. enterocolitica* and *Y. pseudotuberculosis*; bacteria grown at 22–25 °C produce large amounts of O-antigen, while bacteria grown at 37 °C produce only its trace amounts [90]. The loss by *Y. pestis* of the ability to produce LPS with long O-side polysaccharide chains in the course of the evolution turned out to be beneficial for the new lifestyle of *Y. pestis* in a new ecological niche, removing the obstacle that prevented the direct contact of pathogenic factors located on the surface of the bacterial cells with their molecular targets in both the warm-blooded host and flea vector [57].

Thus, the complement-dependent killing of bacteria is among the first lines of defense of mammalian innate immunity against pathogens. The temperature dependence of the manifestation of serum resistance in *Y. pseudotuberculosis* [91] is associated with the synthesis of O-side chains of LPS. The absence of long O-polysaccharides in *Y. pseudotuberculosis* LPS at 37 °C, and in *Y. pestis* also at 20–28 °C, does not prevent the direct contact of AilC with complement resulting in serum resistance [92].

In some Gram-negative enteropathogens, the full-size O-antigen exerts an antiphagocytic role, preventing the uptake of bacteria. Lipooligosaccharide devoid of O-side chains does not protect bacterial cells from being swallowed by host phagocytes. Lipooligosaccharide, which is constitutively produced by *Y. pestis* at any temperature [93] and in *Y. pseudotuberculosis* at 37 °C [94], does not interfere with phagocytosis, but the latter remains incomplete. Viable bacteria within a eukaryotic cell are not recognized by the host’s immune system and are carried through the bloodstream throughout the host’s body.

The loss of O-antigen represents a critical step in the evolution of *Y. pseudotuberculosis* into *Y. pestis* in terms of hijacking APCs, promoting bacterial dissemination and causing the plague. Host cell takeover is guaranteed by the interaction of the core portion of LPS with a DC-specific intercellular adhesion molecule-grabbing non-integrin (DC-SIGN SIGN) (CD209) receptor, expressed by APCs. *Y. pestis* engineered to produce LPS with O-side chains lost its ability to penetrate into the APCs [95]. More recently, it was shown that *Y. pestis* utilizes its core LPS to interact with mouse DC-SIGN-related protein 1 (SIGNR1, CD209b), a C-type lectin receptor expressing on the splenic marginal zone, lymph nodes, peritoneal macrophages and playing a role in lymphocyte migration from the blood into tissues, leading to bacterial dissemination to the lymph nodes, spleen and liver, and the initiation of a systemic infection. It was proposed that the loss of O-antigen represents a critical step in the evolution of *Y. pseudotuberculosis* into *Y. pestis* in terms of hijacking APCs, promoting bacterial dissemination and causing the plague [93]. A highly specific protein–LPS association was found for the R-LPS and the porin proteins of *E. coli, Y. pseudotuberculosis, Y. enterocolitica* and *S. minnesota* co-extracted from these strains as a high molecular weight protein–LPS complex using the phenol–chloroform–petroleum ether method [96,97]. Among bacterial pathogens, representatives of the OmpA family of porin proteins play important roles in adhesion, invasion or intracellular survival as well as the evasion of host defenses or stimulators of pro-inflammatory cytokine production [98]. Unlike in other bacterial pathogens in which OmpA can promote adherence, invasion or serum resistance, the OmpA of *Y. pestis* and *Y. pseudotuberculosis* is restricted to enhancing intracellular survival [99], and we can speculate that this property is LPS dependent.

*Y. pestis* surface protease Pla is among the key pathogenicity factors of the bacterium. The manifestation of the multifunctional activities of Pla requires its binding to LPS. In this case, the spatial proximity of the oligosaccharide of the LPS core of the plague pathogen and the outer loops of the Pla protein molecule makes it possible for LPS to influence Pla conformation. The binding of the core part of LPS to Pla occurs at the amino acid residues Arg_138_ and Arg_171_, which induces conformational changes in the active site, affects its stereometry and modulates the activity of *Y. pestis* plasminogen activator. The substitution of these arginine residues, particularly Arg_138_, reduces the proteolytic activity of Pla and the quantity of Pla in bacteria [92,100,101,102]. The removal of LPS (lipo-chaperone) results in Pla deactivation. The high enzymatic activity of Pla in bacteria without steric interferences generated by the O-polysaccharide side chains is one of the main pathogenetic rewards of the R-form LPS in *Y. pestis*. On the other hand, an alternative suggestion was made that the need for LPS for the manifestation of the activity by the Pla protease ensures the absence of enzymatic activity of the plasminogen activator in the bacterial cytoplasm and its functioning only when incorporated into the outer membrane.

The introduction of *Y. pestis* plasmid pPla that carries the *pla* gene into both the *Y. pseudotuberculosis* strain expressing smooth LPS and its mutant, which, as a wild-type *Y. pestis*, was able to produce only O-antigen-deficient LPS, caused Pla synthesis, export to the outer membrane and processing as in *Y. pestis*. The ability of Pla to activate plasminogen was observed only in the strains producing O-antigen-free lipooligosaccharide [87].

The outer membrane protein X (OmpX)/Ail (adhesion invasion locus) in *Y. pestis* is required for efficient bacterial adherence to and internalization by cultured HEp-2 cells and confers resistance to human and rat serum. The deletion of *ail* in the CO92 strain delayed the time to death by up to 48 h in a mouse model and completely attenuated the virulence of the *ail*-deficient derivate in a rat model of the disease [103].

Working as a single whole, Ail and LPS, through a mutual action, create an optimal Ail conformation on the surface of the bacterial membrane and cause thickening and stiffness of the LPS membrane, which together contribute to the survival of *Y. pestis* in human serum, antibiotic resistance and the integrity of the cell envelope [104].

The Ail from *Y. pestis* and *Y. pseudotuberculosis* is identical, except for one or two amino acids at positions 43 and 126, depending on the *Y. pseudotuberculosis* strain. It was reported that Ail from the *Y. pseudotuberculosis* strain YPIII was not able to bind to host cells. It was found that Ail from both the *Yersinia* species can provide adhesion and invasion, but the long O-side polysaccharide chains of *Y. pseudotuberculosis* LPS prevent the contact of Ail with the host cells [105].

The genetic-engineering modification of the structure of *Y. pestis* LPS has shown the importance of individual components of this biomolecule for bacterial virulence. An isogenic set of knockout mutants [35] based on the full-virulent strain 231 [52], differing in the degree of core truncation, showed that *Y. pestis* knock-out mutants with two or less sugar residues in the LPS core were highly susceptible to antimicrobial cationic peptides and human serum as well as avirulent in murine and guinea pig subcutaneous infection models (Figure 3).

In summary, for the normal functioning of LPS-dependent pathogenicity factors, Pla (bacteria intrahost dissemination), Ail (serum resistance) and antimicrobial cationic peptide resistance, the *Y. pestis* version of this complex macromolecule should include in its core structure at least five to seven sugar residues [35]. All eight wild-type *Y. pestis* core constituent sugars are necessary for the maximal enzymatic activities of Pla.

## 5. Immunogenicity and Protective Efficacy of *Yersinia* Lipopolysaccharide

Immunization with *Y. pestis* LPS alone as well as together with cholera toxin B subunit [67], *Shigella dysenteriae* outer membrane proteins [106] or in a complex with *Y. pestis* outer membrane proteins [107] in all cases resulted in the development of an antibody response to LPS but failed to protect mice against a challenge of virulent *Y. pestis*. Moreover, *Y. pestis* LPS, when injected simultaneously with F1 capsular antigen, produced an immunosuppressive effect in revaccinated *Papio hamadryas* [108].

Although it is not a protective antigen, *Y. pestis* LPS is, nevertheless, of interest for the design of vaccine preparations, since it allows the regulation of the immune response to the introduction of other antigens. Thus, the emergence in the virulent strains of the ability to include the sixth fatty acid residue into its LPS at 37 °C leads to the recognition of the pathogen and prevents its reproduction and, accordingly, avoids the host’s death, but contributes to the formation of intense protective immunity [109]. On the other hand, the inability of a vaccine strain to synthesize endotoxic six-acyl LPS at 28 °C allows the attenuated strain to multiply unhindered and induce approximately the same potent immune response [110].

Priming the innate immune system using the aminoalkyl glucosaminide 4-phosphate (AGP) Toll-like receptor 4 (TLR4) ligands of Toll-like receptor 4 (TLR4), synthetic lipid A mimetics, prolonged the time to death or even protected mice from lethal intranasal challenge with *Y. pestis* CO92. AGP protection was TLR4 dependent and was not observed in TLR4-deficient transgenic mice. AGP therapy, in combination with sub-therapeutic doses of gentamicin, significantly increased survival. It has been suggested that, in combination with other treatments, AGPs could be used to protect immunologically naïve people from the plague [111].

## 6. Phages Target LPS

The exposure of a significant part of the LPS molecule in the extracellular environment makes these parts the main receptors for bacteriophages. The LPS-specific phages of *Y. pseudotuberculosis*/*Y. pestis* are shown in Table 5. Not all the phages of *Y. pestis* use LPS as their only receptor; for example, the receptor of the YpP-R phage extends beyond the LPS core [112], and the outer membrane proteins of *Y. pestis,* Ail and OmpF, participate together with the LPS core in Yep-phi adsorption [113]. YpfΦ can use several different cell surface molecules as receptors [114].

Periodically, papers are published on the isolation of new *Y. pesti*s-specific phages and the study of the spectrum of their host specificity, the possibility of using them for decontamination and therapy. Unfortunately, researchers do not always reach the point of determining the receptors for these bacteriophages [124,125].

## 7. Conclusions

*Y. pseudotuberculosis* is a serologically heterogeneous species, including strains with structurally diverse OPSs, which define the classification of these bacteria into 18 O-serotypes. A peculiar feature of many OPSs is the presence of various of 3,6-dideoxyhexoses, *Y. pseudotuberculosis* being the only bacterial species that produces all known natural isomers of these monosaccharides. OPSs of several O-serotypes contain a unique eight-carbon branched monosaccharide, called yersiniose A (Yer), which has been found in *Y. pseudotuberculosis* for the first time ever. Although diverse, the OPSs can be grouped based on the identical main chain structure or the same terminal side-chain monosaccharide.

In contrast, *Y. pestis* lacks any OPS owing to mutations in 5 of the 17 genes of the O-antigen biosynthesis gene cluster inherited from *Y. pseudotuberculosis.* In the LPS of *Y. pestis* and the R-form LPS of *Y. pseudotuberculosis,* the OPS chain is replaced with a single β-d-GlcNAc residue.

There are significant similarities of the core and lipid A structures of *Y. pseudotuberculosis* and *Y. pestis.* A major difference of *Y. pseudotuberculosis* is the presence of lipid A species that contain a secondary 16:0 acyl group, which are not present in *Y. pestis* due to a mutation in the corresponding acyltransferase gene (*pagP*) [36]. A significant distinction of *Y. pestis* bvv. caucasica and altaica is a shorter core as compared to the main subspecies, *Y. pestis* ssp. *pestis,* due to the inability of the non-main subspecies to incorporate dd-Hep into the LPS.

The LPS structures of both *Y. pseudotuberculosis* and *Y. pestis* undergo essentially the same temperature-dependent variations that contribute to the intrinsic heterogeneity of the LPS and change the charge of the cell surface [5,7,8]. Thus, the elevation of the growth temperature causes a decrease in the positive charge by diminishing the content of Ara4N and an increase in the negative charge by additional phosphorylation in lipid A to produce a diphosphate group.

Several variations, such as the replacement of dd-Hep with Gal in the core and the glycosylation of the phosphate groups of lipid A with Ara4N, are regulated by the two-component signal transduction PhoPQ system [30,32], but this system is not involved in the 3-hydroxylation of Kdo and control of lipid A acylation.

Temperature-dependent LPS structure variations are biologically significant and can be considered as part of a unique mechanism elaborated by *Y. pseudotuberculosis* for adaptation to different conditions in the soil and mammalian host and adopted by *Y. pestis* for the flea-borne type of transmission [5]. Indeed, the incorporation of Ara4N into lipid A, which enhances the resistance of *Y. pestis* to cationic antimicrobial peptides by a decrease in their affinity for the LPS, may promote the growth of *Y. pestis* in fleas, which elaborate antimicrobial peptides as a significant component of their innate immune system. An adaptive response to conditions in the mammalian host includes a decrease in the number of acyl groups in lipid A at 37 °C, which reduces the recognizability of the LPS [32,66] by innate immunity and, as a result, compromises the host’s immune response to infection.

The exact biological role of the core structure variations remains unknown, but it has been speculated that the 3-hydroxylation of Kdo and incorporation of Gal and PEtN to the core at lower temperatures are among the LPS modifications that are beneficial for the asymptomatic persistence of the bacterium in the mammalian host during winter hibernation and in active insects [2,4].

A number of in vitro and in vivo investigations have shown the crucial role of phospholipids and LPSs in the folding and topogenesis of membrane proteins [126,127,128,129,130], as the basis for classifying these lipid-containing molecules as lipo-chaperones.

Summarizing the abovementioned findings, we can state that LPS is not only an endotoxin and a thermostable component of the outer leaflet of the outer membrane of Gram-negative microorganisms, which ensures the structural integrity of the bacterial cell and protects the membrane from aggressive environmental influences, acts as bacteriophage receptor and TLR4 ligand, but is also a kapellmeister, making sure that its orchestra members, protein factors of pathogenicity, played harmoniously and entered on time.

It is obvious that structural studies of LPS, along with studies of the genetic coding of its biosynthesis, are far ahead of studies on the biological significance of this biomolecule, which are in their infancy. It is necessary to make significant efforts to identify the real contribution of various structural elements of different structural variants of this multifunctional molecule, which are actually synthetized within the flea vector or mammalian host, to the molecular mechanisms of pathogenesis and immunogenesis of various clinical forms of plague in various laboratory animals.

There is still a large amount to understand and study, and “microbe hunters” will stay relevant for a long time.

## Figures and Tables

**Figure 1 biomolecules-11-01410-f001:**
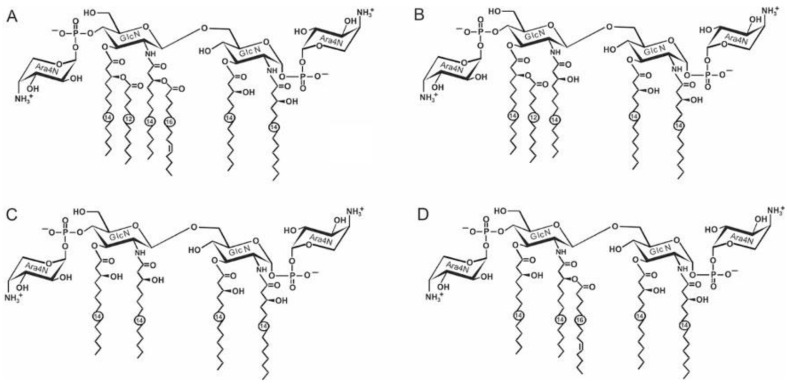
Structures of hexaacyl (**A**), pentaacyl (**B**) and tetraacyl (**C**,**D**) lipid A species of *Y. pestis* and *Y. pseudotuberculosis.* In tetraacyl lipid A of *Y. pestis* (**C**,**D**), Ara4N is present in a non-stoichiometric amount at 37° or completely absent at 6 °C*,* respectively.

**Figure 2 biomolecules-11-01410-f002:**
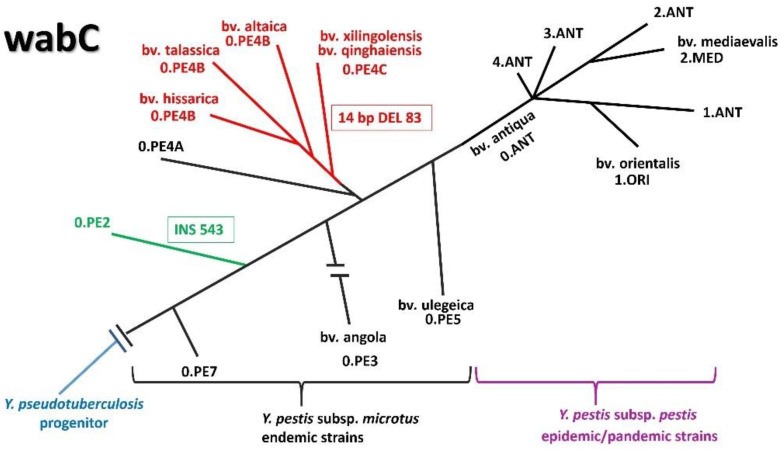
Schematic genomic tree and divergence based on core SNP analysis of 61 *Y. pestis* Genome Assembly and Annotation reports (https://www.ncbi.nlm.nih.gov/genome/browse/#!/prokaryotes/153/, accessed on 23 September 2021). The relationship among subspecies, biovars and SNP types is shown by Kislichkina [52].

**Figure 3 biomolecules-11-01410-f003:**
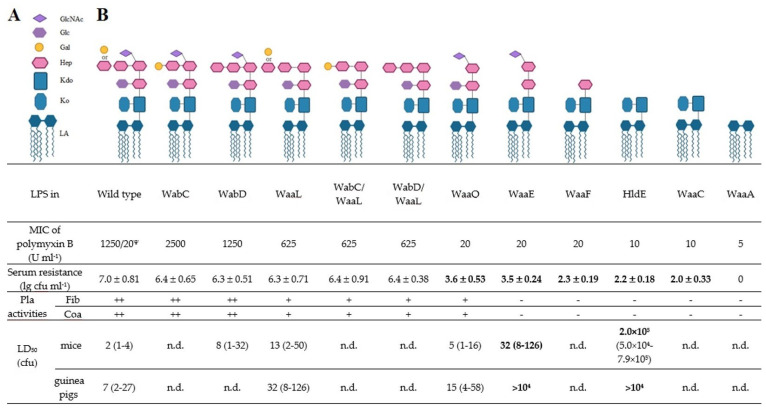
Biological properties of wild-type *Y. pestis* strain 231 and derived LPS mutants. (**A**) Key to LPS moieties. (**B**) LPS core structures of *Y. pestis* 231 and its isogenic mutants. Data are a summary of previous work on LPS of *Y. pestis* 231 [35] LA and wild-type lipid A; when both are present, dd-HepIV and Gal alternate at the non-reducing end. In all strains, terminal Ko is partially replaced with terminal Kdo. Significant changes in biological properties are indicated in bold face. MIC, minimal inhibitory concentration; cfu, colony forming unit; /20^Ψ^, MIC at 6 °C; n.d., not determined; Fib, fibrinolytic activity; a positive fibrinolysis test “++” was represented by complete clot lysis; a positive test “+” was represented by any degree of lysis; a negative test “−” was represented by a solid clot; CoA, coagulase activity; a positive coagulase test “++” was represented by a solid clot; a positive test “+” was represented by any degree of incomplete clotting (from a loose clot to a solid clot in liquid plasma); a negative test “−” was represented by the absence of clotting.

**Table 2 biomolecules-11-01410-t002:** Structures of the core oligosaccharides of *Y. pseudotuberculosis/Y. pestis* cultivated at various temperatures. Substituents present in non-stoichiometric amounts are indicated in italics.

Species, Growth Temperature [Reference]	Major Structure
*Y. pseudotuberculosis/Y. pestis* ssp. *pestis* 37 °C [28] LPS-37	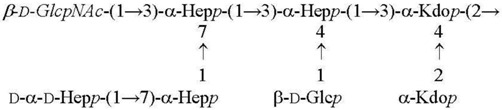
*Y. pestis* ssp*. microti* bvv. caucasica, altaica 37 °C [5] LPS-37	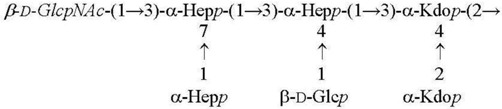
*Y. pseudotuberculosis/Y. pestis* ssp. *pestis* 25 °C [28] LPS-25	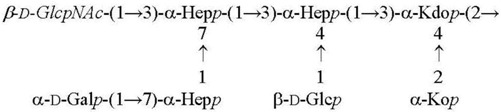
*Y. pestis* ssp. *pestis* 6 °C [29] LPS-6	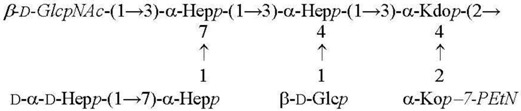

**Table 3 biomolecules-11-01410-t003:** Homologues of *Y. pestis* lipid A biosynthesis and structural modification genes found in the *Y. pseudotuberculosis*/*E. coli* genomes. Sequence conservation relates to alignments with the *Y. pseudotuberculosis* IP 32953 and *E. coli* K-12 sequences.

Gene *Y. pestis* CO92	Related Proteins (% identity *Y. pseudotuberculosis/E. coli)*	Proposed Function
Biosynthesis		
*lpxA*/YPO1056	100% *Y. pseudotuberculosis* CAH22229.1/82% *E. coli* BAA77856.2	UDP-N-acetylglucosamine acetyltransferase
*lpxC*/YPO0561	100% *Y. pseudotuberculosis* CAH19934.1/93% *E. coli* NP_414638.1	UDP-3-O-acyl-N-acetylglucosamine deacetylase
*lpxD*/YPO1054	100% *Y. pseudotuberculosis* CAH22231.1/83% *E. coli* NP_414721.1	UDP-3-O-(3-hydroxymyristoyl)glucosamine N-acyltransferase
*lpxH*/YPO3075	100% *Y. pseudotuberculosis* CAH22231.1/70% *E. coli* NP_415057.1	UDP-2,3-diacylglucosamine diphosphatase
*lpxB/YPO1057*	100% *Y. pseudotuberculosis* CAH20273.1/82% *E. coli* NP_414724.1	Lipid A disaccharide synthase
*lpxK/*YPO1396	100% *Y. pseudotuberculosis* CAH20661.1/70% *E. coli* NP_415435.1	Tetraacyldisaccharide 4′-kinase
*waaA (kdtA)/*YPO0055	100% *Y. pseudotuberculosis* CAH19292.1/80% *E. coli* NP_418090.1	Kdo transferase
*lpxM/*YPO2063	100% *Y. pseudotuberculosis* CAH21284.1/65% *E. coli* NP_416369.1	Lauroyl acyltransferase
*lpxP/*YPO3632	100% *Y. pseudotuberculosis* CAH22835.1/67% *E. coli* NP_416879.4	Palmitoleoyl acyltransferase
Structural Modification		
*pagP/*YPO1744	100% *Y. pseudotuberculosis* CAH20861.1/53% *E. coli* NP_415155.1	Palmitoyltransferase acyltransferase
*arnT/*YPO2418	100% *Y. pseudotuberculosis* CAH21564.1/54% *E. coli* ArnT NP_416760.1	Lipid IV_A_ 4-amino-4-deoxy-L-arabinosyltransferase
*lpxT* (*yeiU*)/YPO1276	99% *Y. pseudotuberculosis* CAH20550.1/62% *E. coli* NP_416679.4	Kdo_2_-lipid A phosphotransferase

**Table 4 biomolecules-11-01410-t004:** Homologues of *Y. pestis* core OS biosynthesis genes found in the *Y. pseudotuberculosis*/*E. coli* genomes. Sequence conservation relates to alignments with the *Y. pseudotuberculosis* IP 32953 and *E. coli* K-12 sequences [35,48,49,50].

Gene *Y. pestis* CO92	Related Proteins (% Identity *Y. pseudotuberculosis/E. coli)*	Proposed Function
CORE BIOSYNTHESIS GENE CLUSTER *waaI*		
*gmhD*/YPO0058	100% *Y. pseudotuberculosis* CAH19295.1/83% *E. coli* BAE77673.1	ADP-l,d-Heptose epimerase
*waaF/*YPO0057	99% *Y. pseudotuberculosis* CAH19294.1/74% *E. coli* NP_418077.1	Heptosyltransferase (HepI)
*waaC*/YPO0056	100% *Y. pseudotuberculosis* CAH19293.1/68% *E. coli* NP_418078.1	Heptosyltransferase (HepII)
*waaA*/YPO0055	100% *Y. pseudotuberculosis* CAH19292.1/80% *E. coli* NP_418090.1	Kdo-transferase (KdoI, KdoII)
*waaE*/YPO054	100% *Y. pseudotuberculosis* CAH19291.1/76% *Serratia marcescens* AAC44433.1	Glycosyltransferase (Glc)
CORE BIOSYNTHESIS GENE CLUSTER *wab*		
*wabC/*YPO0186	99% *Y. pseudotuberculosis* CAH22956.1/61% *Burkholderia pyrrocinia* PXX29563.1	Heptosyltransferase (HepIV)
*wabD*/YPO0187	99% *Y. pseudotuberculosis* CAH22955.1/44% *Proteus mirabilis* MBG3081359.1	Glycosyltransferase (Gal)
CORE BIOSYNTHESIS GENE CLUSTER *waaII*		
*waaQ*/YPO0416	99% *Y. pseudotuberculosis* CAH19795.1/40% *E. coli* NP_418089.1	Heptosyltransferase (HepIII)
*waaL*/YPO0417	100% *Y. pseudotuberculosis* CAH19796.1/59% *Serratia* fonticola WP_074031921.1	O-antigen ligase
structural modification		
*eptB*/YPO4013	99% *Y. pseudotuberculosis* CAH23086.1/63% *E. coli* NP_418002.2	Kdo_2_-lipid A phosphoethanolamine 7″-transferase
*kdoO*/YPO1650	99% *Y. pseudotuberculosis* CAH21657.1/69% *Serratia marcescens* ASL83593.1	Kdo-3-hydroxylase

**Table 5 biomolecules-11-01410-t005:** Components of the *Y. pestis* LPS as a specific receptor for bacteriophages.

Group	Bacteriophage	Receptor	References
T7	PhiA1122	Kdo/Ko of LPS	[115,116,117,118]
	YpP-Y	Hep(I)/Glc of LPS	[112,119,120]
	YpP-R	Beyond the LPS core	[119,120]
	Yep-phi	LPS; OmpF; Ail	[113,121]
	Pokrovskaya (Yepe2,YpP-G)	Hep(II)/Hep(III) of LPS	[120,121,122]
	T7_Yp_	Hep(I)/Glc of LPS	[112]
P2	L-413C	GlcNAc of LPS	[112,119,120]
	P2 *vir1*	GlcNAc of LPS	[112]
T4	PhiJA1	Kdo/Ko of LPS	[112,119,120]
	YpsP-PST	Hep(II)/Hep(III) of LPS	[112,119,120,123]

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
