# Peer review of "Lipopolysaccharide of the Yersinia pseudotuberculosis Complex"

_biomolecules, 2021, doi:10.3390/biom11101410_

Round 1
Reviewer 1 Report
Comments:
In this manuscript, Knirel et al. did comprehensive summarization about studies in Y. pseudotuberculosis LPS.
Concerns:
- The manuscript needs significant editing for proper English syntax and diction.
- Lines 52-54, I am wondering that how O8, O13 and O14 are designated if no O-units are present in those Yersinia strains.
- Please give a full name for the GlcNAc or GalNAc abbreviation that is firstly shown in the manuscript.
- Lines 142-143, penta-acylated lipid A not tera-acylated lipid A was shown in Fig. 1D.
- Lines 151-152, the sentence is not clear.
- Lines 156-157, the grammar of the sentence is not correct. Nine enzymes locate in the cytoplasm….not “the pathway”. Replace “takes place” by “locate”.
- Line 215, “temperature: not “temperatures”
- Lines 240-248, this paragraph is not clear.
- Line 346, the sentence is not proper.
- Lines 416-417, absence of the lpxL gene in Y. pestis is the most important for evasion of innate-immune surveillance. Usually, the pagP is an inducible gene in most Gram-negative bacteria. So, I guess the pagP gene has been mutated in Y. pestis may provide an addition strategy to avoid immune recognition during evolution.
- Lines 276-277, You have mentioned that “LPS is an integral component of the outer membrane that is in direct contact with the host organism, taking part in the protection of bacterial cells against complement system components and other host defense mechanisms and systems”. Y. pseudotuberculosis synthesizes smooth LPS at 26 °C and rough LPS at 37 °C.
Later on in lines 429-431, you have mentioned that “Y. pseudotuberculosis, which is completely resistant to the bactericidal action of serum when cultured at 37 °C, but susceptible at 26 °C”
Can you add some explanations about how Y. pseudotuberculosis that is variable dependent on temperatures interacts with the complement system in mammalian hosts?
- Lines 466-468, need to clarify differences of the LPS in this paragraph to the next paragraph.
Also, altered LPS charge also can affect Pla activity (Sun et al. 2013; PMCID: PMC3639600).
Reviewer 2 Report
This manuscript by Knirel and coauthors presents a comprehensive review of lipopolysaccharide of Yersinia pseudotuberculosis and Y. pestis. This group has a vast experience in this field and presented review covers several aspects related to LPS structural variations related to impact of temperature on lipid A and O-specific polysaccharide, non-stoichiometric modifications, LPS recognition by innate immune system and role of LPS in virulence. Authors also cover certain findings with virulence factors such as surface protease Pla and OmpX protein in relation to LPS. It is a balanced review, however certain aspects could be added in this review, which are missing as listed below.
- Only minor information is presented in this review related to transcriptional regulation of LPS biosynthesis and its structural alterations. Authors should add information about impact of alternate sigma factors, two-component systems, post-transcriptional control of LPS.
- LPS transport (Lpt proteins) system is not described. Conservation of Lpt proteins could be added.
Technical issues: A. Authors mention in lines 173-175 “As with E. coli, the addition of……strictly necessary…”. It is a misleading statement and authors must correct it. Kdo-independent late acylation in the lipid A has been shown in E. coli under slow growth conditions at low temperatures or upon overexpression of genes encoding late-acyl transferases (Gorzelak et al Int J Mol Sci. 2021 ijms 22105099, Reynolds and Raetz; Biochemistry 2009, 48, 9627–9640, Klein et al J. Biol. Chem. 2009, 284, 15369-15389). These references should be added and the text and statement modified.
B. Line 191: A complete inner core is required------. Do you mean Kdo incorporation or something else in the inner core. Also, it would be more appropriate to use Ara4N incorporation rather than binding.
Modification of sentences: Text needs several modifications as many statements are not clear and written in confounded manner. Just few examples:
- Lines 25-26: Its lipid A moiety called lipid A—remove redundancy and modify
- Line 112: “awake” mammals- modify
- Line 132: “highly acylated-----could be precise such as hexaacylated
- Lines 144-145: there are present lipid A ---please modify
- Line 374: (PEtN) also occasionally modified -could be modified
- Line 375: PEtN “embedding” should be incorporation
- Line 473: introduction of carrying-----should be modified
- Line 484 “as a whole unit”….please change.
Round 2
Reviewer 2 Report
It is now improved, confusing sentences changed and better edited. Nice review